# DEEP GENERATIVE MODELING FOR IDENTIFICATION OF NOISY, NON-STATIONARY DYNAMICAL SYSTEMS

## ABSTRACT

A significant challenge in many fields of science and engineering is making sense of time-dependent measurement data by recovering governing equations in the form of differential equations. We focus on finding parsimonious ordinary differential equation (ODE) models for nonlinear, noisy, and non-autonomous dynamical systems and propose a machine learning method for data-driven system identification. While many methods tackle noisy and limited data, non-stationarity – where differential equation parameters change over time – has received less attention. Our method, dynamic SINDy, combines variational inference with SINDy (sparse identification of nonlinear dynamics) to model time-varying coefficients of sparse ODEs. This framework allows for uncertainty quantification of ODE coefficients, expanding on previous methods for autonomous systems. These coefficients are then interpreted as latent variables and added to the system to obtain an autonomous dynamical model. We validate our approach using synthetic data, including nonlinear oscillators and the Lorenz system, and apply it to neuronal activity data from C. elegans. Dynamic SINDy uncovers a global nonlinear model, showing it can handle real, noisy, and chaotic datasets. We aim to apply our method to a variety of problems, specifically dynamic systems with complex time-dependent parameters.

## 1 INTRODUCTION

Many fields of science and engineering now benefit from unprecedented amounts of data due to increased efforts and technological breakthroughs in data collection. The challenge is to use these measurements to expand our understanding of dynamical systems in areas like climate science, neuroscience, ecology, finance, and epidemiology. Machine learning methods, such as neural networks, are widely used for data-driven modeling, offering high prediction accuracy but limited interpretability. In contrast, traditional techniques that identify ordinary and partial differential equations (ODEs and PDEs) provide interpretable and generalizable insights into the system's underlying physics. While neural networks may lose accuracy as conditions change, in many systems the governing differential equations remain reliable. The key question is whether we can combine the strengths of deep learning with the clarity and simplicity of data-driven differential equation models.

A key challenge in data-driven system identification is that many systems exhibit nonlinear behavior, such as switching between dynamical regimes (30; 17; 21; 34). These "hybrid systems" (33), where continuous dynamics shift at discrete events, are more challenging to define and simulate than classical systems with smooth vector fields (1; 5). Standard methods often assume that the data comes from a system governed by a fixed set of equations and terms, but time-varying hidden variables can further hinder identification of the system's underlying dynamics. This motivates our focus on non-autonomous (or non-stationary) systems, where sudden shifts or hidden continuous dynamics complicate accurate modeling and prediction.

We introduce dynamic SINDy, a data-driven method for finding non-autonomous dynamic systems with switching or continuously-varying latent variables. These systems are described by:

$$\dot{\mathbf{x}} = f(\mathbf{x}(t), t) \tag{1}$$

where $\mathbf{x}$ is vector-valued. A simple such example is $\dot{\mathbf{x}} = A(t)\mathbf{x}$. Importantly, we focus on systems where the time-varying component and the main variables of interest $\mathbf{x}$ are separable (e.g., $\dot{\mathbf{x}} =$

$f(\mathbf{x}, t) = \sin(t)\mathbf{x}$, but not $\dot{\mathbf{x}} = f(\mathbf{x}, t) = \sin(t\mathbf{x})$. Another assumption is that if multiple trajectories of the system are available, these all display the same underlying switching or hidden variable dynamics.

Dynamic SINDy combines the interpretability of differential equations with the power of deep learning. It uses a deep generative model to uncover sparse governing equations directly from data, employing a variational autoencoder (VAE) to generate time series for differential equation coefficients. This enables data-driven discovery of equations for noisy and non-autonomous systems. The paper is organized as follows: Section 2 introduces key concepts, including SINDy, variational autoencoders, and dynamic VAEs. Section 3 describes the methodology, covering the datasets and the dynamic SINDy framework. Section 4 demonstrates dynamic SINDy's performance on various systems, including non-autonomous oscillators, Lorenz, Lotka-Volterra, and neural activity data from C. elegans. It also compares dynamic SINDy to switching linear dynamical systems (41) and group sparse regression methods (19). Section 5 concludes the paper.

## 2 BACKGROUND AND PREVIOUS WORK

### 2.1 SYSTEM IDENTIFICATION OF NON-LINEAR DYNAMICAL SYSTEMS (SINDY)

SINDy (Sparse Identification of Nonlinear Dynamics) (43) is a data-driven method that uses sparse regression on a library of nonlinear candidate functions to match data snapshots with their derivatives, revealing the governing equations. The method assumes that only a few key terms explain the system's dynamics. More specifically, consider $\mathbf{x}(t) \in \mathbb{R}^d$ governed by the ODE: $\dot{\mathbf{x}}(t) = f(\mathbf{x}(t))$. Given $m$ snapshots of the system $\mathbf{X} = [\mathbf{x}(t_1), \mathbf{x}(t_2), ..., \mathbf{x}(t_m)]^T$ and the estimated time derivatives $\dot{\mathbf{X}} = [\dot{\mathbf{x}}(t_1), \dot{\mathbf{x}}(t_2), ..., \dot{\mathbf{x}}(t_m)]^T$, we construct a library of candidate functions $\Theta(\mathbf{X}) = [1, \mathbf{X}, \mathbf{X}^2, ..., \mathbf{X}^p, \sin(\mathbf{X}), \cos(\mathbf{X}), ...]$. We then solve a sparse regression problem, $\dot{\mathbf{X}} = \Theta(\mathbf{X})\Xi$, to identify the optimal coefficients $\Xi$ and to reduce the number of terms, enforcing parsimony. A sparsity-promoting regularization function $R$ is added to the final loss to yield:

$$\hat{\Xi} = \mathrm{argmin}_{\Xi}(\dot{\mathbf{X}} - \Theta(\mathbf{X})\Xi)^2 + R(\Xi) \tag{2}$$

Several innovations have followed the original formulation of SINDy (42; 18; 32; 14). For instance, integral and weak formulations (47; 20) have enhanced the algorithm's robustness to noise. Of relevance to our study, SINDy's generalization to non-autonomous dynamical systems has been previously explored using group sparsity norms (42) or clustering algorithms (33).

### 2.2 (DYNAMIC) VARIATIONAL AUTOENCODERS FOR SYSTEM IDENTIFICATION

The Variational Autoencoder (VAE) (35; 11) combines neural network-based autoencoders with variational inference for probabilistic modeling and data generation. Unlike standard autoencoders, VAEs stand out due to two key features: (i) VAEs encode input data $X$ as a distribution in the latent space, allowing the decoder to generate new data by sampling from this distribution; and (ii) a regularization term ensures the latent space resembles a standard (e.g., normal) distribution, making it continuous (nearby points generate similar outputs) and complete (all points produce meaningful data). Further mathematical details can be found in Supplementary Material (SM) Section 1.1. A related method of interest is HyperSINDy (28). It combines VAEs with SINDy to discover differential equations from data. The VAE approximates the probability distribution of equation coefficients, so that once trained, HyperSINDy generates accurate stochastic dynamics and quantifies uncertainty, making it a powerful tool for model discovery.

In order to adapt the VAE/SINDy framework to non-autonomous systems, we would like to implement generative architectures that capture the temporal dependencies in sequential data. Dynamic VAEs (DVAEs) is an approach that extends VAEs to handle time series data (24). A number of DVAE architectures are described that use recurrent neural networks or state-space models to address both latent and temporal relationships (36; 37; 13; 22; 23; 2; 26; 39; 46). We specifically use timeVAE (10), which has shown strong performance in generating time series data by processing entire sequences with dense and convolutional layers to capture correlations. Our approach is flexible, allowing the VAE architecture to be swapped for other models better suited to the data or system under study (SM Sec. 1.2).

### 2.3 OTHER MACHINE LEARNING METHODS FOR NON-AUTONOMOUS DYNAMICAL SYSTEMS

Traditionally, methods for handling hybrid or switching systems often involve dividing time or space into segments (16). For instance, reduced-order models for nonlinear systems segment time intervals into smaller windows, then build a local, reduced approximation space for each segment (25; 6; 27). Clustering methods are also employed for modeling, particularly in complex fluid flows, where clusters represent states that can transition via a Markov model (12; 3) or via dynamic mode decomposition with control (31; 4). Through data-informed geometry learning, authors in (48) reconstruct the relevant "normal forms", which are prototypical realizations of the dynamics, providing bifurcation diagram and insights about the parameters even for non-autonomous systems. Yet another method (49) applies Koopman operator theory using DMD algorithms to find time-dependent eigenvalues, eigenfunctions, and modes in linear non-autonomous systems.

We compare dynamic SINDy with two existing methods (Section 4.6). First, we look at a method (recurrent SLDS) (41) that extends switching linear dynamical systems (SLDS) (15; 9) by generating transitions through changes in a continuous latent state and external inputs, rather than relying on a discrete Markov model for switching states. This model breaks the data into simpler segments and is interpretable, generative, and efficiently fitted using modular Bayesian inference. Second, we examine a method from (15; 9) that uses group-sparse penalization for model selection and parameter estimation. This method assumes shared sparsity across parameters by applying group-sparsity regularization to smaller time windows in the data, identifying the system for each segment, and then combining the results.

## 3 METHODS

### 3.1 DATASETS

We use a synthetic dataset capturing dynamics of a non-autonomous harmonic oscillator:

$$\dot{x} = A(t)y$$
$$\dot{y} = B(t)x \tag{3}$$

where $A(t)$ and $B(t)$ are the time-varying coefficients of the ODE. The time dependence of these coefficients renders the system non-autonomous and difficult to discover using classical methods. We test our approach to see if it can handle switching coefficients, as well as explore continuously varying coefficients, such as sinusoidal functions at different frequencies or finite Fourier series (Figure 1A, Suppl. Fig. 2). To ensure robustness against randomness, we add Gaussian noise with varying levels of variance to the time series.

We replace a set of constant coefficients with a set of time series (sigmoidal, switching, sinusoidal, finite Fourier series) for more complex systems, such as the chaotic Lorenz system:

$$\dot{x} = \sigma(t)(y - x)$$
$$\dot{y} = x(\rho(t) - z) - y \tag{4}$$
$$\dot{z} = xy - \beta(t)z$$

We use large-scale neural recordings from whole-brain imaging to model neuronal population dynamics. C. elegans, with its 302 precisely mapped neurons, offers an ideal balance of simple behavior and complex neuronal activity. We analyze calcium imaging data from Kato et al., which includes neural recordings from the head ganglia and manual annotations of seven behaviors: forward movement, reversal, two types of reversal-to-forward turns, and two forward-to-reversal turns (38). Previous studies show that high-dimensional neuronal activity simplifies into low-dimensional patterns, with clear clusters in principal component space representing forward and backward movements. This provides a valuable opportunity to study the link between neural activity and behavior.

### 3.2 SYSTEM IDENTIFICATION FOR NON-AUTONOMOUS DYNAMICAL SYSTEMS

We explore various VAE architectures designed for inference and generation of time series data. The input is the original time series $X$, and the output are time series of ODE coefficients:

$$\Xi_{1:t} = V(\mathbf{X}_{1:t}) \tag{5}$$

where $V$ is the (VAE) architecture, and $\Xi_{1:t}$ is the output time series. 'Autoencoder" is a misnomer because the input is not designed to match the output in this VAE architecture. The ODE coefficients are linearly combined with a pre-determined SINDy library of basis functions to yield $\hat{\dot{\mathbf{X}}}$:

$$\hat{\dot{\mathbf{X}}}(t) = \Theta(\mathbf{X}(t), t) \cdot \Xi(t) \tag{6}$$

where $\Theta(\mathbf{X}(t), t)$ is a row vector comprising of a polynomial basis up to cubic monomials: $\Theta(\mathbf{X}(t), t) = [1 \quad X_1(t) \quad ... \quad X_n(t) \quad X_1^2(t) \quad ... \quad X_n^3(t)]$, where $X_i$ are features of $\mathbf{X}$. Although we choose a polynomial basis for all of our experiments, the basis can change depending on the problem at hand or any prior information (43).

Our goal is to match $\tilde{\dot{\mathbf{X}}}$, the derivative we estimate from data using numerical methods, to the output $\hat{\dot{\mathbf{X}}}$ of our model (Eq. (5-6)). The loss function takes the following form:

$$\text{loss} = \sum_t ||\tilde{\dot{\mathbf{X}}}(t) - \hat{\dot{\mathbf{X}}}(t)|| + \lambda_1 R_{kld} + \lambda_2 R(\Xi) \tag{7}$$

where $\lambda_{1,2}$ are hyperparameters of the optimization and $R, R_{kld}$ are regularization terms. $R_{kld}$ is the Kullback-Leibler divergence (KLD) loss, part of the ELBO (evidence lower bound) loss in VAEs (see SM Section 1.1). Regularization terms impose that $\Xi(t)$ is sparse (in coefficients) to encourage parsimony and that $\Xi(t)$ has minimal total variation. More details about the loss function and training, specifically the inference and generation models, can be found in the SM, Sec. 1.3.

We focus on two neural network architectures in our experiments. First, timeVAE (SM Sec. 1.2.1, Suppl. Fig. 1A) is simple for proof-of-concept testing (10); however, its major drawback is that it requires the entire time series as input, which can be impractical for long sequences, especially in high-dimensional systems due to memory constraints. To address this, we introduce a new architecture called dynamic HyperSINDy (SM Sec. 1.2.2, Suppl. Fig. 1B). Alternatively, we can use DVAE architectures, which allow sequential data input, overcoming timeVAE's limitations (24).

## 4 RESULTS

### 4.1 SYSTEM IDENTIFICATION OF NON-AUTONOMOUS HARMONIC OSCILLATORS

We begin by identifying noisy, non-autonomous dynamical systems using a simple toy model – a non-autonomous harmonic oscillator with time-varying ODE coefficients (Eq. 3, Figure 1). First, we vary the coefficient $A(t)$ in a switch-like fashion (Figure 2(a)-(c)). The system behaves like a classic harmonic oscillator, but with a frequency switch. The inferred coefficients (Figure 2, Suppl. Fig. 4) and the reconstructed trajectories (Suppl. Fig. 6) align well with the true values. These trajectories are generated during testing, with $z$ sampled from a standard normal distribution.

When varying both $A(t)$ and $B(t)$ as sinusoids with different frequencies, the resulting trajectories generally capture their oscillations, though some higher error and a large outlier appear toward the end (Figure 2(d), Suppl. Fig. 3). We also successfully reproduce coefficients composed of multiple frequencies (a finite Fourier series) in Figures 2(e)-(f). In (f), some error occurs in the first half because the system approaches a fixed point where the derivative is nearly zero. In such cases, system identification becomes difficult, as multiple solutions can produce the same dynamics.

### 4.2 UNCERTAINTY QUANTIFICATION IN NON-AUTONOMOUS HARMONIC OSCILLATORS

We use VAEs to quantify uncertainty by estimating the standard deviation of the coefficients over time. Therefore we generate multiple trajectories by sampling $z$ from a standard normal distribution during testing. Figures 3(a)-(c) show examples of trajectories from networks trained on noisy data with two noise levels: low (0.01) and high (0.5) standard deviations. As expected, trajectories vary more under high noise than low noise. Our results show that the estimated standard deviation generally follows the true coefficient variations. First, we compute the standard deviation across generated samples at each time point and average these deviations (Figure 3 B(a)). Second, we subtract a smooth mean from the trajectory samples and calculate the standard deviation over time (Figure 3 B(b)). Both methods demonstrate that standard deviation aligns with the ground truth, particularly for switch signal coefficients, but is less clear for Fourier series coefficients. Further work is needed to improve standard deviation estimation, considering the VAE architecture and hyperparameters.

A

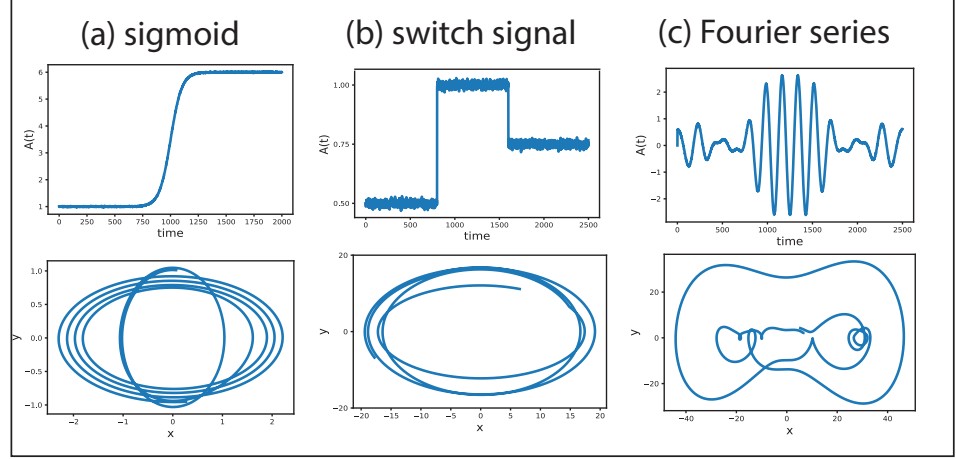

B

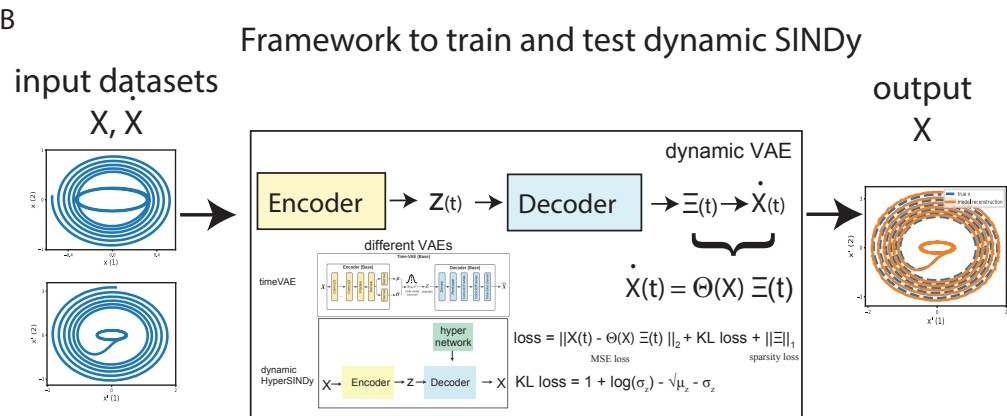

Figure 1: (A). Synthetic dataset to test dynamic SINDy with non-autonomous harmonic oscillators (Eq. (3)). Top: Example (SINDy) coefficient time series $A(t)$; Bottom: corresponding trajectories in phase space (B). Dynamic SINDy general architecture schematic; two DVAEs shown as example.

### 4.3 System identification in a non-autonomous, chaotic toy dataset

We next modified the Lorenz system by allowing one of its key parameters ($\sigma, \rho, \beta$) to vary over time, similar to the non-autonomous harmonic oscillator examples. The modified Lorenz equations are:

$$\dot{x} = \sigma(t)(y - x)$$
$$\dot{y} = x(\rho - z) - y$$
$$\dot{z} = xy - \beta z \tag{8}$$

Here, $\sigma(t)$ varies over time as a sigmoid, switch function, sinusoid, or as a Fourier series with 7 overlapping frequencies. Despite these changes, the system still converges to a global attractor.

For system identification, we used two dynamic SINDy architectures: the timeVAE, effective for shorter time series (1000–2000 points), and dynamic HyperSINDy (SM Sec. 1.2.2), suitable for longer time series. Training occurs in two stages: first, we apply a sparsity penalty to set small coefficients to zero; second, we fine-tune the remaining coefficients. After training, we remove the encoder and generate time series from the decoder, closely matching the ground truth across different parameters and functions (Figure 4, Suppl. Fig. 5). Hyperparameters are listed in SM, Sec. 1.3.

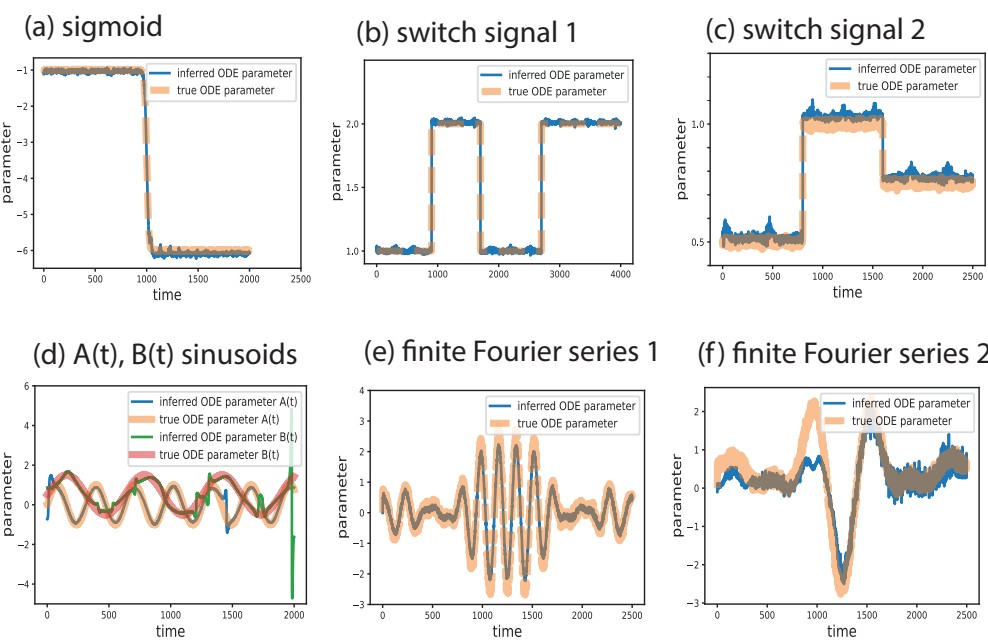

Figure 2: Dynamic SINDy generates coefficient time series that match ground truth for non-autonomous harmonic oscillators (Eq. (3)). (a)-(f) different examples of time-varying $A(t)$, $B(t)$.

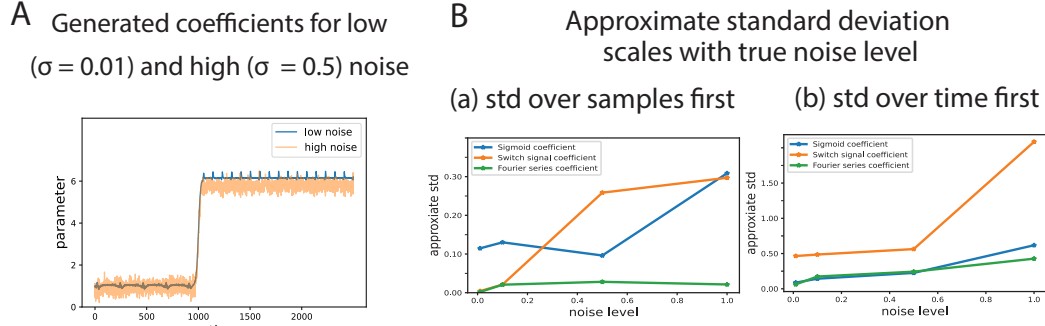

Figure 3: (A) Dynamic SINDy generates coefficient time series for different levels of Gaussian noise in the coefficient. (B) Inferred noise (standard deviation, or std) scales with ground truth Gaussian noise for different time-varying coefficients. (a) std computed over many generated samples, then averaged (b) std computed over time, then averaged over samples (see Sec. 4.2)

### 4.4 DYNAMIC SINDY USED FOR IDENTIFYING LATENT VARIABLES AND THEIR DYNAMICS

Dynamic SINDy is particularly useful for discovering hidden (latent) variables from incomplete datasets. We demonstrate this using a toy model from ecology: the Lotka-Volterra equations, which describe predator-prey dynamics between two species:

$$\dot{x} = \alpha x - \beta xy$$
$$\dot{y} = -\gamma y + \delta xy \quad (9)$$

In our example, we only observe the prey population, $x$, and aim to use dynamic SINDy to uncover the hidden predator population, $y$, and reconstruct a full 2D autonomous system in $x$ and $y$.

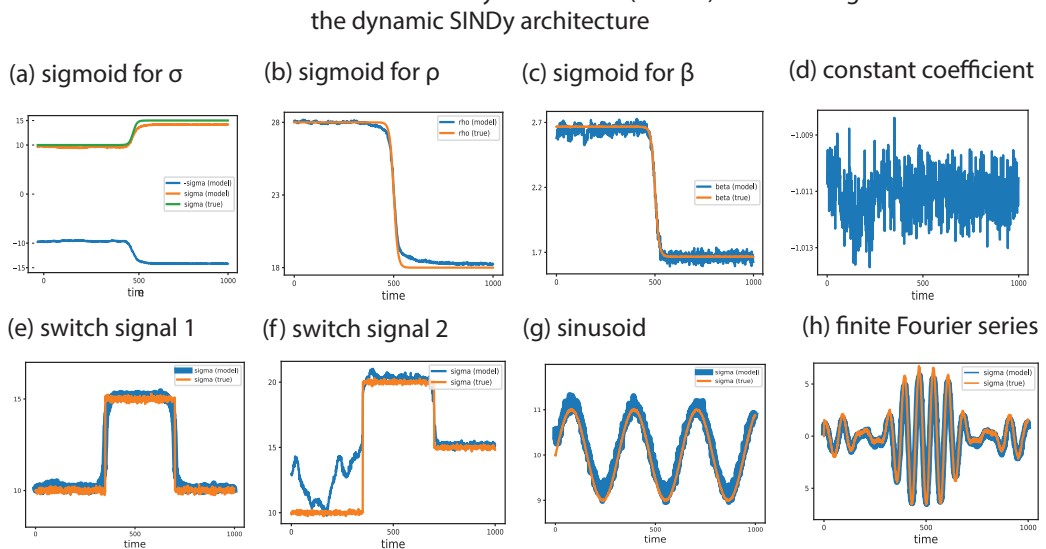

Figure 4: Dynamic SINDy generates coefficient time series that match ground truth for Lorenz dynamics (Eq. (8)). (a)-(h) different examples of time-varying $\sigma(t)$, $\rho(t)$, $\beta(t)$.

We apply dynamic SINDy to $x$, using a library with just three terms: $x, x^2, x^3$. As expected, $\dot{x}$ is expressed solely in terms of $x$, with the $x^2$ and $x^3$ terms vanishing. We derive a time series for the coefficient $\tilde{y}$, where $\dot{x} = x\tilde{y}(t)$. This inferred $\tilde{y}$ correlates with the hidden $y$, where $\tilde{y} = q \cdot (\alpha - \beta y)$, with $q$ being a scaling factor applied to $x$ before using dynamic SINDy. From $\tilde{y}(t)$, we can infer $y$ and compare it to the true population. In noiseless data, we accurately reconstruct the predator dynamics 6A, but with more noise, recovery becomes harder 6B. Using $\tilde{y}$, we form a new 2D system of equations:

$$\dot{x} = a \cdot x\tilde{y}$$
$$\dot{\tilde{y}} = b + c \cdot x + d \cdot \tilde{y} + e \cdot x\tilde{y} \tag{10}$$

where $a, b, c, d, e$ are new model parameters. Comparing the inferred coefficients to the original Lotka-Volterra system by changing variables from $y$ to $\tilde{y}$ and using standard SINDy and the pysindy package, we find a close match (Figure 6C). We applied this same approach to the non-autonomous harmonic oscillator (Eq. 3, SM Sec. 2.1), further confirming that dynamic SINDy can successfully identify hidden variables and form complete autonomous systems.

### 4.5 Dynamics in the nematode C. elegans during locomotion behavior

Modern neuroscientific data is noisy, nonlinear, and incomplete, with recordings from hundreds or thousands of neurons, yet many network features and neurons remain unmeasured. This makes it a challenging test for dynamic SINDy's ability to model such complex systems. To demonstrate our method's potential, we use a dataset of C. elegans neural activity (Sec. 3.1, (38)). Unlike previous approaches that rely on probabilistic state space models (40) or hidden Markov models (44; 45; 7), our method uncovers a global nonlinear switching model (8; 29). This model captures key features of the neural data: two stable fixed points representing forward and reversal behaviors, transitions between them, and variability in those transitions, reflecting real neural dynamics.

We first apply PCA to the data from one animal to obtain low-dimensional dynamics that cluster according to behavioral states (Figure 6A,B). Notably, only two dimensions are necessary to differentiate between forward, backward, and turning behaviors, although differentiating between various types of turns requires more dimensions. For a minimally complex model, we focus on the neural trajectory described by the dominant PC mode and its derivative. Our goal is to identify a

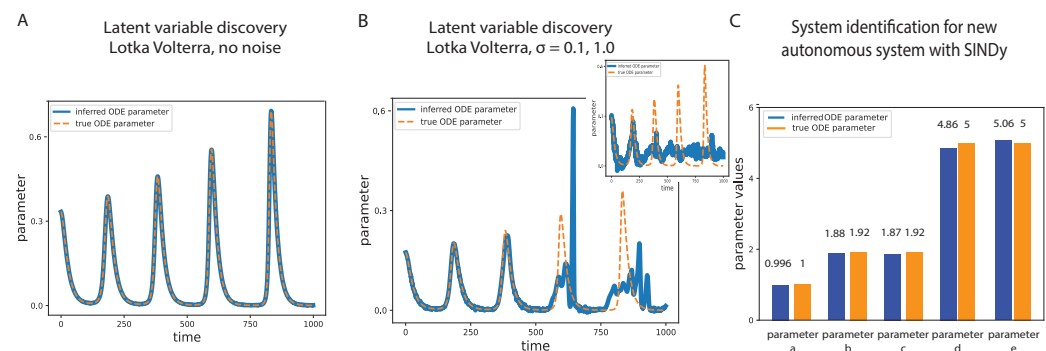

Figure 5: Dynamic SINDy can be used for latent variable discovery. (A). Inferred (blue) versus true (orange) $y$ time series, from noiseless Lotka-Volterra. (B). Same as in (A), when noise of different standard deviations $\sigma$ is added to the Lotka-Volterra trajectory. (C). Inferred versus true coefficients in the Lotka-Volterra 2D ODE system, using SINDy for system identification.

nonlinear, parsimonious, and global model of the form:

$$\dot{x} = y \tag{11}$$
$$\dot{y} = f(x, \beta) + u(t) \tag{12}$$

where $x$ is the data projected onto the first principal component, $f$ is an unknown function, $u(t)$ is a potential switching or control signal, and $\beta$ is a vector of parameters we would like to fit to our data.

We apply dynamic SINDy to minimize the error between the model's derivatives $(\dot{x}(t), \dot{y}(t))$ and the dominant PC derivatives from the data. Following sections 4.1 and 4.3, we identify the sparsity pattern of the SINDy coefficients, enforcing $\dot{x} = y$. The method highlights the terms $1, x, y, x^2, x^3$ for describing $\dot{y}$ and calculates their time-varying coefficients (SM Sec. 3.1, 3.2). To further simplify the model, we set coefficients for all variables to be constant, except the flexible term $u$ which we can also reduce to a time series of switches (Figure 6D, see SM Sec. 3.2.1) without meaningfully affecting global dynamics. Converting $u$ into a switching signal simplifies this term, helping to regularize the model and improve interpretability. This approach aligns with previous studies showing bistability and sudden transitions in behavior.

Our approach identifies a cubic function for the differential equation model: $\ddot{x} = f(x, \beta) + u(t) = 0.002 \cdot x^3 + 0.0087x^2 - 0.22 \cdot y + 0.05 \cdot x + u_i$, where $u_i$ alternates between $u_0 = -0.266$ and $u_1 = 0.044$ (see SM Sec. 3.1 for details). Each time $u$ switches, the cubic function shifts, altering the fixed point that the trajectory converges to. This switching signal $u$ enables the transitions between the two fixed points, which correspond to forward and reversal behaviors. Overall, the reconstructed data captures key features like fixed points and transitions (Figures 6E-F, 6G). By labeling the trajectory based on behaviors, we align the inferred dynamics with the training data (Figure 6H).

The reconstructions are accurate regardless of whether we use the processed switching term $u$ (Figure 6 D) or the original time series $u$ (Figure 6 C). However, $u$ alone does not adequately explain the data; removing other terms leads to poor fits or instability. By systematically eliminating different terms, we find that all are essential for capturing the dynamics. When we initialize the inferred ODE system from different starting points and use $u$ from training, the resulting dynamics qualitatively match the data. This suggests that our method effectively avoids overfitting. Unlike Morrison et. al., which relies on selecting a model based on human-labeled behavioral states, dynamic SINDy is fully data-driven and does not require labeled data to partition the phase space (29). Additionally, unlike Fieseler et. al., our model accommodates nonlinear dynamics with two stable fixed points (8). A key advantage of our ODE model is the potential for biologically interpretable parameters (see (29). SM Sec. 3.3 offers a more comprehensive discussion of the benefits of our framework, comparing our global nonlinear switching model to previous studies. In summary, we have demonstrated that dynamic SINDy can do automatic data-driven model discovery, generating a nonlinear model with minimal input from the data scientist.

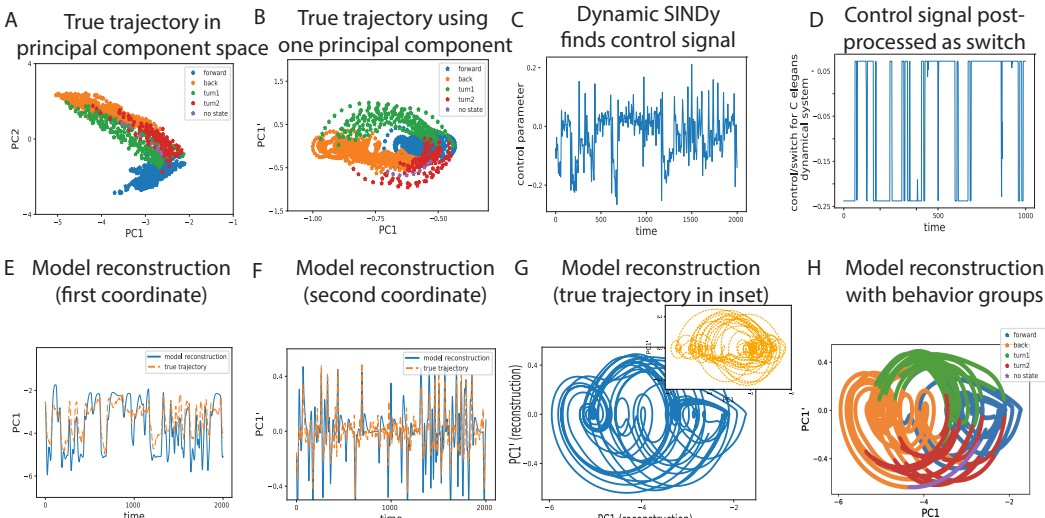

Figure 6: (A). C. elegans neuronal activity is low-dimensional and clusters according to behavior; (B). Neuronal activity in phase space given by the first principal coordinate and its derivative; (C). Dynamic SINDy inferred constant term; (D). Processing coefficient in (C) as switch; (E) and (F) ODE model's match to ground truth trajectories; (G) (and (H)) 2D model trajectory (with labeled behavior).

## 4.6 DYNAMIC SINDY AND OTHER METHODS FOR SYSTEM IDENTIFICATION

We begin by comparing dynamic SINDy with switching linear dynamic systems (SLDS) (15; 9) and its extension, rSLDS (41). SLDS uses a discrete latent variable, $z_t$, to partition the state space between switches (see SM Sec. 4.1), simplifying complex nonlinear dynamics into more manageable linear segments. The rLDS extension allows switches to depend on continuous latent states and external inputs using logistic regression (41). We evaluate how well SLDS/rSLDS identifies switching signals in the dynamical systems studied so far, specifically inferring where the latent variable $z$ changes for switching to occur. We use coefficients based on sigmoids and two types of switching signals (refer to the "ground truth" in Figure 7A). Running SLDS or rLDS generates samples of the latent variable $z$ that segment the training trajectory, enabling us to compare this segmentation with the actual ground truth switches.

We find that for a sigmoidally varying coefficient, SLDS identifies the switch fairly well (Figure 7A (a), (b)), as shown by the colored trajectories and the insets comparing the $z_t$ time series to the ground truth; although for Lorenz dynamics, the predicted change in the latent state $z_t$ is slightly delayed relative to the actual switch (Figure 7A (b)). However, SLDS struggles when there are multiple state switches in the time series (Figures 7A (c) and (d)). For the harmonic oscillator, rSLDS produces a model with too many switches and is more complex than the ground truth. For Lorenz dynamics, both SLDS and rSLDS switch periodically whenever the dynamics change between attractors, but this periodicity does not match the true switches defined by the coefficients. To address these challenges, we added time as a new dimension to the dataset, represented as a simple feature vector $[1, 2, \ldots, T]$, where $T$ is the total number of time steps. The goal was for SLDS/rSLDS to recognize that the switches are time-dependent rather than state-dependent. However, this addition did not improve the performance of SLDS or rSLDS.

Another method for identifying non-autonomous systems, discussed in references (42; 19) and SM Sec. 4.2, involves dividing the trajectory into smaller time windows and applying SINDy to each segment while enforcing a consistent sparsity pattern across all windows. We tested this approach on two toy datasets, using SINDy coefficients modeled as sigmoids, sinusoidal functions, and a Fourier series with seven frequencies. Without the group sparsity regularization, the sparsity patterns varied across the windows, highlighting the importance of group sparsity for achieving a coherent solution. The group sparsity approach worked well when the coefficients were sigmoid

functions with varying smoothness (Figure 7 B (a) and (c)). However, it struggled with sinusoidal and Fourier series coefficients, particularly in the Lorenz system (7 B(b) and (d)). In cases of misidentified coefficients, the algorithm also generated incorrect sparsity patterns. We conclude that neither SLDS or rSLDS, nor the group sparsity method are as effective as our method in identifying non-autonomous dynamical systems from data.

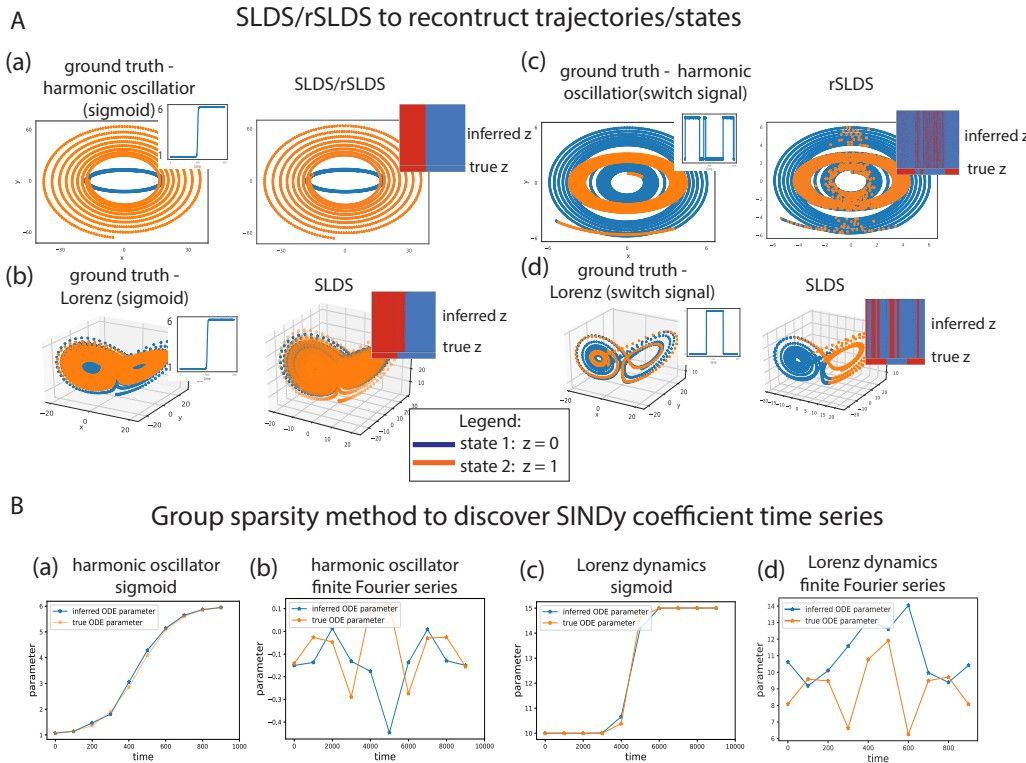

Figure 7: (A) SLDS/rSLDS infers switching behavior for non-autonomous harmonic oscillators and Lorenz dynamics as coefficients vary. (a)-(d) left: ground truth dynamics, labeled switch values colored blue and orange. Inset shows true coefficient. (a)-(d) right: dynamics labeled by inferred switch. Inset: ground truth $z$ and samples of discrete latent values $z$ labeled by switch. (B) Inferred SINDy coefficients versus ground truth using group sparsity method.

## 5 CONCLUSION

We have developed dynamic SINDy, an extension of SINDy designed for data-driven identification of noisy, nonlinear, and non-autonomous dynamical systems, as well as for discovering latent variables. We demonstrated the effectiveness of dynamic SINDy on both benchmark synthetic datasets and a real, noisy, chaotic dataset of neuronal activity from C. elegans.

However, our method has some limitations. First, the DVAE architecture has many hyper-parameters to tune, and results may not be robust to these settings, especially in noisy datasets. A systematic approach for hyperparameter tuning and addressing multiple solutions is necessary. To prevent overfitting, we should encourage simpler time series through regularization. Although we included a simple approximation of total variation term in our loss function, realistic datasets might require more sophisticated regularization. Future research should explore different DVAE architectures to evaluate their accuracy in reproducing dynamics and their ability to quantify uncertainty. It would also be valuable to apply our method to experiments with different trial dynamics and types of noise than those studied here (e.g., varying switching points across trials). Lastly, we aim to apply dynamic SINDy to realistic data from fields like biology, physics, and engineering to uncover hidden dynamics and fully utilize its potential for discovering latent variables.

## 6 REPRODUCIBILITY STATEMENT

A detailed list of models, neural network architectures, algorithms, parameters, hyperparameters, etc. can be found in the Supplementary Material. The methods section in the main text contains important information about the synthetic datasets we have created to test our model, as well as a description of the C. elegans dataset from (38). A set of useful Python scripts is provided. While the code is still *messy*, the authors commit to improve it and make it quite accessible. The hope is that it will soon become an important reference to accompany the manuscript.

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
