# OpenReview forum: "Deep Generative Modeling for Identification of Noisy, Non-Stationary Dynamical Systems"
_ICLR.cc/2025/Conference — ICLR 2025 Conference Withdrawn Submission_

### Official Review · Reviewer_Mq5x · 2024-11-02

**Soundness:** 3
**Presentation:** 3
**Contribution:** 3
**Rating:** 6
**Confidence:** 4

**Summary:**

This paper presents dynamic SINDy, a machine learning method designed to identify the governing equations of noisy, nonlinear, and non-autonomous dynamical systems from data. Dynamic SINDy combines variational inference with sparse identification of nonlinear dynamics to identify time-varying coefficients in sparse ordinary differential equations. The method is particularly valuable for non-stationary systems with changing parameters over time. The contributions include
- Modeling Time-Varying Coefficients: Dynamic SINDy employs deep generative models, specifically variational autoencoders (VAEs), to learn the time-varying nature of ODE coefficients. This enables the identification of non-autonomous systems that exhibit complex dynamics.
- Uncertainty Quantification: The use of VAEs allows dynamic SINDy to quantify uncertainty in the estimated ODE coefficients. This is crucial for understanding the reliability of the identified model.

**Strengths:**

- Modeling Time-Varying Coefficients: Dynamic SINDy employs deep generative models, specifically variational autoencoders (VAEs), to learn the time-varying nature of ODE coefficients. This enables the identification of non-autonomous systems that exhibit complex dynamics.
- Uncertainty Quantification: The use of VAEs allows dynamic SINDy to quantify uncertainty in the estimated ODE coefficients.
- Latent Variable Discovery: Dynamic SINDy can effectively uncover hidden (latent) variables that influence the observed dynamics. This is demonstrated through an example with the Lotka-Volterra equations, where only prey population data is available.
- Application to Real-World Data: The paper validates dynamic SINDy's capabilities on both synthetic and real-world data, i.e. the C. elegans data.

**Weaknesses:**

- Uncertainty Quantification: though sec 4.2 shows the estimated standard deviation follows the truth, it only confirms the accuracy of estimation. Nonetheless, it is unclear what use the uncertainty could provide.
- The examples are mostly using systems that driven by mean/input processes. It's unclear how the proposed method would perform for noise driven dynamics.
- In the C. elegans example, the assumed form depends heavily on prior knowledge (dimensionality, input) on the dynamics. Though the proposed method has shown very good performance, it's unclear what scientific insights the proposed method could offer especially for the systems that people have limited knowledge about.

**Questions:**

- What can one do with the uncertainty? Is it necessary for accurate estimate? Would the uncertainty provide insights for scientific questions? Suggestion: elaborate more or showcase scenarios where the uncertainty is useful vs. method.
- How does the method perform for noise driven dynamics? e.g. system with multiple meta-stable points or line attractors.
- What scientific insights the proposed method could offer for the systems that don't have particular a priori form of ODEs? If we don't know u(t) switches, would it discover that?
- Line 461 typo: rLDS
- Suggestion: put Dynamics SINDy reconstruction together with SLDS result in Fig. 7.

---

### Official Review · Reviewer_zDkz · 2024-11-03

**Soundness:** 2
**Presentation:** 3
**Contribution:** 2
**Rating:** 3
**Confidence:** 4

**Summary:**

The authors propose DynamicSINDY, an approach that aims to learn sparse ODEs with time-varying coefficients from noisy, non-stationary time series data. The authors achieve this by combining SINDy with sequential VAEs to probabilistically infer the coefficients and their time-varying values. The method is evaluated on three synthetic datasets and a calcium imaging dataset of C. elegans.

**Strengths:**

- The authors address an important and often overlooked problem in modeling time series data: learning interpretable models from non-stationary dynamical systems.
- The problem and the proposed method are presented clearly, and the paper is generally easy to follow.

**Weaknesses:**

- While the motivation of the problem is clear, I find that the conducted experiments fail to convince the reader of the method's impact in real-world settings. The experiments mainly focus on synthetic datasets that are artificially created to fit this problem and avoid most challenges often encountered in real-world datasets (high dimensionality, non-Gaussian noise, large space of possible coefficients). I think the paper would be greatly benefit if the authors can demonstrate the method in such contexts.

- The C. elegans dataset used to demonstrate the method is quite simple, and the method is only applied to low-dimensional representations obtained via PCA (which in this case is enough to explain the data). The impact of using DynamicSINDY in this case is not well motivated, and the obtained results don't add any much scientific insights, especially since identifiability is not discussed.

- While this is not necessarily always a weakness, the proposed method is a straightforward combination of two existing approaches. Taken together with the limited experiments section and the lack of significant technical innovations, I find the overall contribution of the paper in its current form rather limited.

**Questions:**

- Variational inference is known to provide overconfident uncertainty estimates because of the KL term encouraging mode-covering behavior. Can the authors discuss this further and the impact of this on the method?
- How robust are the identified parameters? Is there a quantitative relationship between the robustness and (1) the size of the library, (2) the level of noise in the system?
- It is stated that the method can only deal with non-stationarity arising from separable time-varying variables. Can the authors elaborate more on why this is the case?

---

### Official Review · Reviewer_NQwk · 2024-11-03

**Soundness:** 3
**Presentation:** 2
**Contribution:** 2
**Rating:** 5
**Confidence:** 4

**Summary:**

the authors introduce dynamic sindy — an extension of the sindy for uncovering nonstationary dynamical systems from data.  the authors use a time-series VAE architecture to map their data to time-varying coefficients that are linearly combined with a fixed library of basis functions to produce an estimate of the data derivative.   they conduct experiments using several toy datasets showing that their dynamic sindy can recover the coefficients of time-varying dynamical systems, even in the case that the entire system is not observed.  finally, they show on low-dimensional representations of c. elegans neural recordings, that their method  recovers a representative dynamical system of the first principal component.

**Strengths:**

dynamical system reconstruction or system identification is an important topics, and learning more interpretable models of system dynamics, such as time varying ode representations like the authors, has broad application to many fields.  additionally, the authors consider a comprehensive amount of toy experiments — i appreciate that the authors considered experiments with several time varying motifs (i.e. switching, sigmoid, switch, fourier) and show that their method can recover the time varying coefficients with a calibrated measure of uncertainty.

**Weaknesses:**

i found the separation between what has previously been done in the literature and what are the authors exact main novel contributions to be unclear; for example, more precise statements about the differences with hypersindy (around line 096) would have been very helpful.  in its current form, it is hard to parse from the manuscript what their exact technical advances are.

a lot of real-estate in the paper is taken up by the experimental plots.  often i found the amount of information conveyed by the plots disproportionate to the amount of space they take up — making more compact figures seems like it would work to the authors advantage.  additionally, information could be conveyed better i.e. thick lines and their ordering (i.e. green/blue lines in Fig 2d are not clear).  Fig 1B has a lot of small labels and the zoomed out view of timeVAE does not feel like it helps much.

**Questions:**

have the authors applied their method to any datasets requiring a higher dimensional latent space to see how quality of the learned dynamical system scales with dimensionality of the latent space?

have the authors considered comparisons to a method more adept at handling more smooth like transitions of dynamics such as [1] which considers a smoothly switching latent system.

[1] Kurle, Richard, et al. "Deep rao-blackwellised particle filters for time series forecasting." Advances in Neural Information Processing Systems 33 (2020): 15371-15382.

---

### Official Review · Reviewer_sBqV · 2024-11-04

**Soundness:** 3
**Presentation:** 2
**Contribution:** 3
**Rating:** 5
**Confidence:** 4

**Summary:**

The paper proposes ‘dynamic sparse identification of nonlinear dynamics’ (dynamic SINDy), a deep learning framework for identifying governing equations in noisy, non-stationary and nonlinear dynamical systems (DS). By combining variational autoencoders (VAEs) and previous work on SINDy, it enables unsupervised inference of the underlying ODE systems’ parameters while extracting a global and parsimonious nonlinear dynamical model. The approach is validated on both synthetic and real-world data and is compared to other methods in the field, demonstrating great potential for scientific machine learning community.

**Strengths:**

- Learning a parsimonious representation of non-autonomous DS is extremely important and relevant in many scientific disciplines, which makes the approach very promising.
- I think it is highly interesting that the encoder-decoder architecture is able to predict the ODE parameters with this level of fidelity in an unsupervised fashion (as there is no direct reconstruction loss for the ODE parameter time series involved in the loss function (7)).
- The method is tested against other baselines and also on a real-world dataset (C. elegans).

**Weaknesses:**

- The authors should stick to the ICLR style guide and use the 'author, year' reference style instead of mere numerical numbers (i.e. APA style instead of IEEE). This increases readability and helps the reader to understand the train of thought of the authors, as one directly sees on which work the authors base certain statements.
- Center box in Fig. 1B is in parts hard to read as (font) sizes vary a lot. I think it would be better to shrink down Fig. 1A a touch and to increase size of Fig. 1B, especially as it describes the main framework of the manuscript.
- I also think the figure group titles ('suptitles') above Fig. 1, 2, and 4 are superfluous and their message should be put into the figure caption. This would create additional space (e.g. to compensate for the change in referencing style).
- I think ‘dynamic HyperSINDy’ deserves a bit more attention in the main text, which lacks explanation on how this approach really works. Explaining this method in the supplement makes the corresponding results a bit hard to read and almost forces the reader to read the supplement section 1.2.2.
- All of the employed (benchmark) datasets are fairly low dimensional (2-3D). The authors do not address the scalability of the method to high dimensional systems (which can not be sufficiently described by the first few PCA components). I think this is a major drawback, as this setting is highly relevant to many real-world systems.

Minor details:
- typo: Fig. 3B y-axis label say “approxiate std”
- l. 353 It just says 6A and 6B, while the authors probably reference Fig. 5A and 5B? Also l. 360 it says 6C instead of 5C.
- typo: supplement l. 262 it says weight decay of 1e5 (I assume 1e-5?)

**Questions:**

- For the switch signals (Fig. 2 a-c, also Fig. 3A low noise setting), the inferred ODE parameter time series seem to exhibit high frequency oscillations on top of the correct switch-like dynamics. Is there an intuitive explanation why the encoder-decoder architecture struggles in inferring the correct switching dynamics and how this could be addressed?
- Results of Fig. 3B look rather weak to me, can the authors report Pearson’s $r$ of noise lvl vs. std?
- I’m confused by section 4.6 & Fig. 7; How exactly does the dynamic SINDy approach compare now to the proposed baseline methods based on SLDS and (vanilla?) SINDy with a group sparsity norm? I think Fig. 7 would be much clearer if the authors would find a design to compare all comparison methods side-by-side.
- ll. 409-411: Can the authors provide references for the mentioned studies?
- How do other methods like reservoir computing compare to the dynamic SINDy approach qualitatively and quantitatively in the settings discussed in the manuscript (see e.g. [1])?
- How does the approach perform on e.g. benchmarks used in [2], which exhibit different bifurcations than the ones discussed in this paper?

I am very happy to increase my score if the authors adequately address my concerns and questions.

References:

[1] Köglmayr, Daniel, and Christoph Räth. "Extrapolating tipping points and simulating non-stationary dynamics of complex systems using efficient machine learning." Scientific Reports 14.1 (2024): 507.

[2] Patel, Dhruvit, and Edward Ott. "Using machine learning to anticipate tipping points and extrapolate to post-tipping dynamics of non-stationary dynamical systems." Chaos: An Interdisciplinary Journal of Nonlinear Science 33.2 (2023).

---

### Note · Authors · 2024-11-25

**Comment:**

We thank the reviewers for their thoughtful comments and the time they spent reading our manuscript.
We have decided it is on our best interest right now to withdraw our paper.

best regards,
-the authors

**Withdrawal Confirmation:**

I have read and agree with the venue's withdrawal policy on behalf of myself and my co-authors.